# In-vitro engineered human cerebral tissues mimic pathological circuit disturbances in 3D

Aref Saberi [1,2✉], Albert P. Aldenkamp[3,4,5], Nicholas A. Kurniawan [1,2✉] & Carlijn V. C. Bouten [1,2]

In-vitro modeling of brain network disorders such as epilepsy remains a major challenge. A critical step is to develop an experimental approach that enables recapitulation of in-vivo-like three-dimensional functional complexity while allowing local modulation of the neuronal networks. Here, by promoting matrix-supported active cell reaggregation, we engineered multiregional cerebral tissues with intact 3D neuronal networks and functional inter-connectivity characteristic of brain networks. Furthermore, using a multi-chambered tissue-culture chip, we show that our separated but interconnected cerebral tissues can mimic neuropathological signatures such as the propagation of epileptiform discharges.

[1] Department of Biomedical Engineering, Eindhoven University of Technology, Eindhoven, the Netherlands. [2] Institute for Complex Molecular Systems, Eindhoven, the Netherlands. [3] School for Mental Health and Neuroscience, Maastricht University Medical Center, Maastricht, the Netherlands. [4] Department of Behavioral Sciences, Epilepsy Center Kempenhaeghe, Heeze, the Netherlands. [5] Department of Electrical Engineering, Eindhoven University of Technology, Eindhoven, the Netherlands. ✉email: a.saberi.aref@gmail.com; kurniawan@tue.nl

Three-dimensional (3D) neuronal models, such as brain organoids, combined with recent advances in high-resolution 3D imaging techniques, gene editing, single-cell omics, and patient-derived induced pluripotent stem cells (iPSCs), have provided pioneering platforms for understanding various aspects of brain development[1–5] and brain pathologies[6–14]. Furthermore, fused region-specific organoids and assembloids[3,15–17] have been successfully developed to recapitulate in-vivo inter-regional and intercellular interactions in 3D. However, despite this exciting progress, building a manipulatable in-vitro model to study the altered or disrupted 3D functional interconnectivity in multiregional network pathologies such as a focal epileptic seizure remains a major challenge[18,19].

Efforts have been made to create engineered platforms for studying interconnectivity between 3D neuronal cell cultures (e.g., using interconnected spheroids[20–23]). However, these approaches lack either the 3D connectivity between the interconnected co-cultures (since the connections are guided through micro channels) or the cellular diversity and complex functionality of organoid approaches (for an overview, see recent reviews by Brofiga et al.[18] and Park et al.[19]). There is a clear need for a new approach to develop neuronal tissue models that retain the in-vivo biomimicry potential of organoids, while presenting the possibility of spatial control of the tissue configuration in a well-defined engineered culture platform that allows 3D connectivity[19,24].

To meet this need, here we introduce a novel culturing method to develop human cerebral tissues via matrix-supported active (migrative) reaggregation of cells (MARC). Unlike other protocols reported for generating brain organoids, which use a quick, mechanically-enforced aggregation of the dissociated cells, the formation of the 3D tissue (hereinafter referred to as "cerebral tissue") in this study is initiated by active reaggregation of the cells during chemically-induced differentiation, with the immediate 3D extracellular support of Matrigel (Fig. 1a; see also experimental details in Methods).

## Results

To promote cellular reaggregation and engineering of multi-regional cerebral tissues, we designed a neuronal differentiation protocol employing phased introduction and withdrawal of the culture additives used for neuronal tissue patterning. After ~80% confluence of human-induced pluripotent stem cell (hiPSC) culture under feeder-free conditions, neuronal induction was initiated through the introduction of SMAD inhibitors (dorsomorphin and SB431542), a GSK-3 inhibitor (CHIR99021), SHH, and b-FGF. After 7 days of induction, the cells were enzymatically dissociated and homogenously resuspended in a mixture of Matrigel and neural differentiation medium containing b-FGF, SHH, and FGF8 (this step is henceforth considered Day 0 of the MARC culture). Thence, neural differentiation took place in a 3D environment for 7 days, accompanied by the rapid formation of spheroids with a size of 200–300 μm (Fig. 1b, Day 7). Pre-terminal differentiation started when b-FGF was removed from the medium, resulting in the formation of neurite outgrowth and bundles connecting the spheroids to each other (Fig. 1b, arrows and arrowheads). These spheroids merged over time (~2 weeks), resulting in large cerebral tissues with a size of 2–4 millimeters (Fig. 1b, Day 15, 20). Finally, SHH and FGF8 were withdrawn and the cultures were kept in the maintenance medium. The cerebral tissues continued growing during culture for 90 days and expressed markers of distinct progenitor and mature neuronal (Glutamatergic, GABAergic, and dopaminergic) and neuroglial (astrocytes and oligodendrocytes) identities (Fig. 1c, d and Supplementary Fig. 1). The observed cell-type diversity and multiregionality in the MARC-produced cerebral tissues were comparable to those found in cerebral organoids obtained using standard SFEBq protocol (Supplementary Fig. 1).

To test the neuronal functionality of MARC-produced cerebral tissues, we evaluated the neuronal interconnectivity within the intact 3D tissues using live calcium imaging. We observed that cerebral tissues at age 4 weeks exhibited extensive spontaneous and synchronized calcium surges throughout the tissues (Fig. 2a, b and Supplementary Movie 1). To quantify this population-wide intercellular synchronized activity, we computed the pairwise linear correlation coefficient $r$ from the intensity time-trace of 387 regions-of-interest (ROIs) representing detected single neurons in the cerebral tissues. The correlation matrix between all ROI pairs shows that the majority of the ROI pairs had low $r$-values, but a substantial number of pairs were highly correlated (Fig. 2c). We defined two ROIs to be functionally connected when $r > 0.6$, following Eguiluz et al.[25]. A functional neuronal connection was found for 304 pairs (~0.4% of all ROI pairs analyzed) and depicted in a spatial connection map (Fig. 2d and Supplementary Data 1), demonstrating a functional neuronal network.

Our data show increased synchrony between the nodes with a higher amount of connections (Fig. 2e). These observations are consistent with the topological features of scale-free networks, which have been suggested to be important for synchronized functional networks[26–28]. The distribution of the number of connections follows a power law with a decay constant of $-2$ (Fig. 2f), consistent with the characteristics of a scale-free network[29] and in agreement with clinical measurements of whole-brain activity[30]. The presence of a small number of hyper-connected "hub-like" cells (up to >20 connections in our case) and a large number of cells with few connections results in a low average number of connections (~1.5 in our cerebral tissues). It has been proposed that the brain achieves large-scale interconnectivity between brain regions despite the low average number of connections through a modular network topology, whereby the network is composed of subnetworks ("modules") of densely interconnected neurons ("nodes")[31,32]. To assess the network modularity in our cerebral tissues, we analyzed the functional connectivity using the iterative Louvain community-detection algorithm[33,34] (see Methods). The algorithm identified 3 distinct modules within the tissue with sparse intermodular node connections (Fig. 2g, h). Moreover, each module includes its own local hubs that are highly interconnected (inset in Fig. 2h), which ensure global integration of the functional interconnections across the overall network[35]. Within each module, the nodes are interconnected in a hierarchical topology, from a few hub nodes with a high number of connections that are closely connected to each other to peripheral nodes at the outer edges of the network topology (Fig. 2i). Taken together, the analysis demonstrates rich interconnectivity in our cerebral tissues reminiscent of the emerging attributes of functional networks in the brain.

Next, we sought to demonstrate the utility of MARC-produced cerebral tissues to model a neurological disorder affecting network interconnectivity, such as epilepsy, in a controlled in-vitro environment. Physiologically, epileptic seizures are characterized by a transient occurrence of abnormal excessive or synchronous neuronal discharges and spatial propagation of these abnormal discharges. To experimentally investigate the propagation of epileptiform discharges between cerebral tissues, we designed and fabricated a polydimethylsiloxane (PDMS) chip that allows for the generation of two cerebral tissues in separate and individually-accessible chambers which are connected through a porous membrane (iS3CC chip; see Supplementary Fig. 2). The design and dimensions of the vertically-tapered chambers were uniquely chosen to suit the one-pot formation of

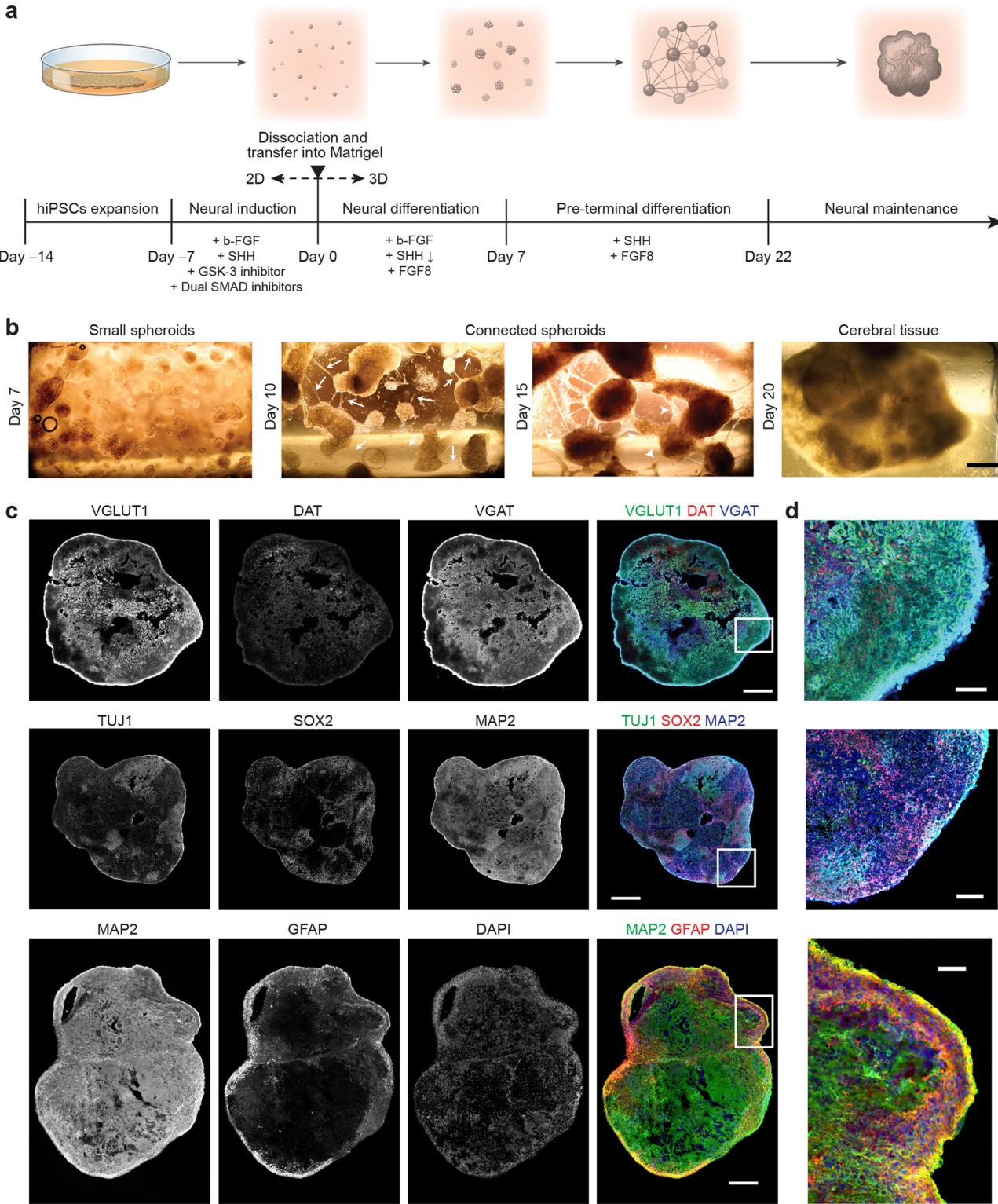

**Fig. 1 Formation of human cerebral tissues via matrix-supported active reaggregation of cells (MARC). a** Schematics of the MARC culture method, showing the different culture steps. Timeline and additives supplied in each step are indicated. **b** Example phase-contrast images at the different 3D-culture phases, showing the matrix-supported active reaggregation of the cells into cerebral tissues. Single dissociated cells suspended in Matrigel grew into small spheroids (Day 1–7). During pre-terminal differentiation, neurite outgrowths extended from the spheroids (white arrows), and merged into neurite bundles (white arrowheads) between spheroids (Day 10, 15). The spheroids migrated and subsequently merged into large cerebral tissues (Day 20) ($n = 120$ samples across six independent experiments). Scale bar: 500 µm. **c** Immunohistochemical co-staining of cryosections of MARC-produced cerebral tissues at Day 90 revealed the presence of markers of neural progenitor cells (NPCs; SOX2), early and mature neurons (Tuj1 and MAP2), mature excitatory Glutamatergic neurons (VGLUT1), inhibitory GABAergic neurons (VGAT), mature dopaminergic neurons (DAT), and astrocytes (GFAP), indicating the cellular diversity of the MARC-produced cerebral tissues ($n = 5$ samples across two independent experiments). Scale bar: 500 µm. **d** Zoom-ins of the images in **c** as indicated by the white squares. Scale bar: 100 µm.

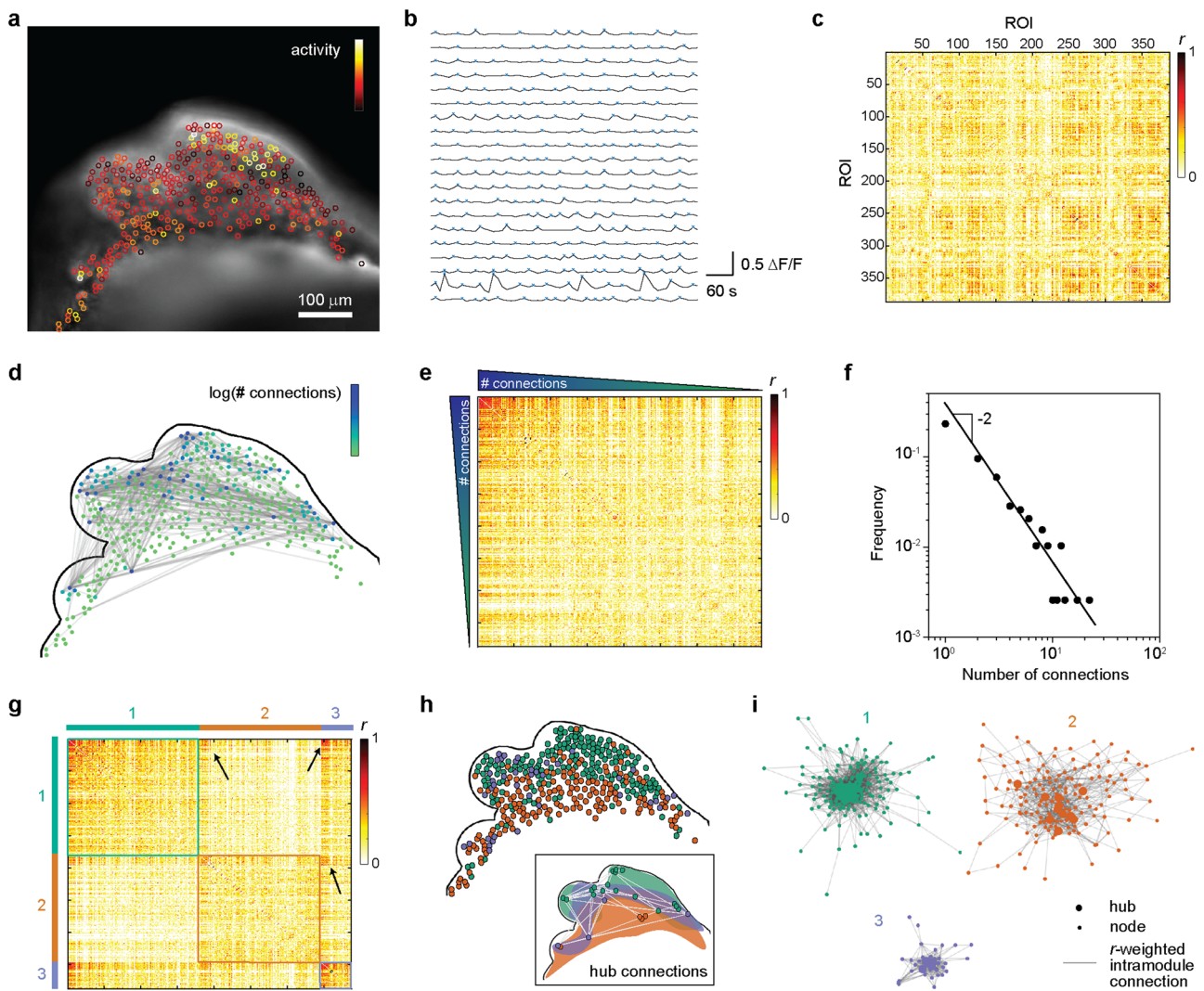

**Fig. 2 Functional network interconnectivity in intact MARC-produced cerebral tissues. a** Neuronal activity in a cerebral tissue at week 4 of MARC culture. A snapshot of live fluorescence calcium imaging on the intact cerebral tissue is overlaid with regions-of-interest (ROIs) color-coded based on the frequency of detected calcium surges ("activity") in each ROI. See Supplementary Movie 1 for the corresponding time-lapse imaging data. Scale bar: 100 μm. **b** Representative time traces of normalized intensity (ΔF/F) from 20 ROIs. Blue crosses indicate detected transient spikes. Time traces from 387 ROIs in the image are used to compute the correlation coefficient $r_{ij}$ between ROI pairs $i$ and $j$ (see Methods for details of calculation). **c** Correlation matrix between any pair of the 387 ROIs, showing the $r$-value for each pair. **d** Depiction of functional connectivity network in the cerebral tissue. The contour of the cerebral tissue, corresponding to **a**, is shown. Two ROIs are defined to be functionally connected when $r > 0.6$ and shown as gray lines in the connectivity map. Further, each ROI in the connectivity map is color-coded based on the number of functional connections it has. **e** The same correlation matrix as in **c**, but with the ROIs sorted based on the number of functional connections they have. The high density of ROI pairs with a high number of connections and high $r$-value suggests a non-random network topology. **f** Distribution of the number of functional connections. The distribution follows a power law with a decaying power of −2, demonstrating a scale-free cerebral tissue functional network. **g** Clustered correlation matrix. To test whether the network exhibits modular topology, the functional connectivities are analyzed using the Louvain algorithm[33], which indicates that the network contains three communities/modules. The correlation matrix in **d** is then reordered so that the nodes in the same module (color-coded with 1, 2, and 3) are positioned together. The high density of cross-module node pairs with high $r$-values (arrows) suggests the existence of hub connections between modules. **h** The localization of the nodes in the three modules. The color of the nodes corresponds to the color-coding of the three modules in **g**. The inset shows the spatial regions that enclose the nodes identified in the three modules, together with the hub nodes (defined as nodes with a number of intra-module connections larger than the 90th percentile in the module) and the cross-module hub connections (gray lines). **i** Topological representation of the intra-module functional connectivity networks. To illustrate the topological proximity of highly connected nodes, each module network is shown using the Fruchterman–Reingold algorithm[44], where the length of the lines connecting nodes is proportional to $1 − r$ (i.e., short lines indicate a high correlation coefficient between the node pairs and the converse). The hubs in each module are also indicated. The central positioning of the hub nodes, as well as the close topological proximity between the hub nodes, highlight their status as intramodular connector nodes in the cerebral tissue functional network.

MARC-produced cerebral tissues and to maintain nutrition and oxygen supply, while simultaneously allowing visualization of both chambers in a side-by-side configuration (Fig. 3a, d). The membrane separating the chambers had pore sizes of 8 μm to keep cerebral tissues separated for individual treatments, yet allow spontaneous neurite interconnection across the membrane. As anticipated, the cerebral tissues generated using the MARC method in the two chambers were observed to spontaneously form functional connections with each other through the porous membrane as a result of spheroid

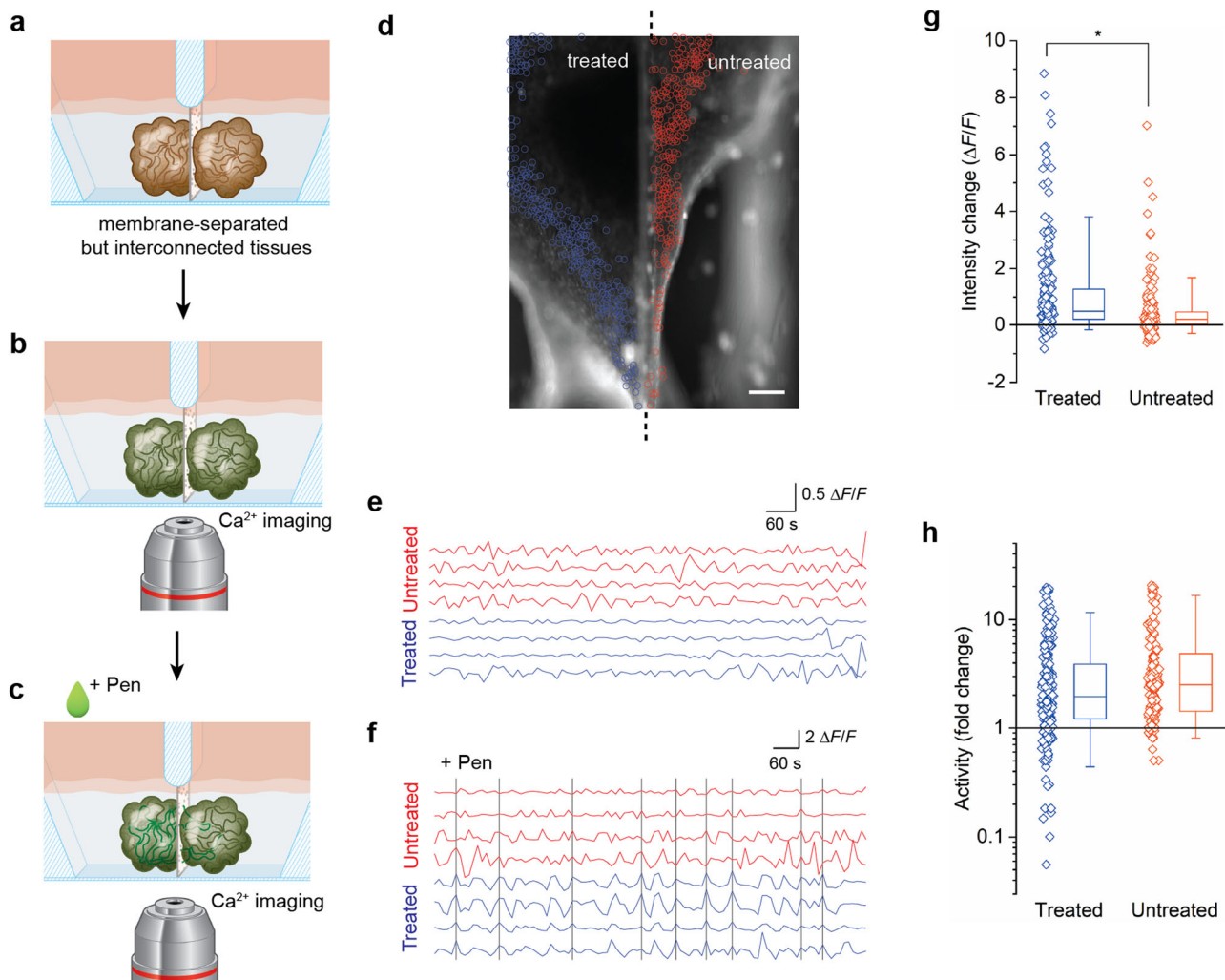

**Fig. 3 Signal propagation between interconnected cerebral tissues. a–c** Schematics indicating the steps involved in the Penicillin-treatment experiments. Two cerebral tissues were generated in the two chambers of the PDMS-based iS3CC chip (see also Supplementary Fig. 2 for more details of the chip) and formed a connection through the porous membrane (**a**, see also Supplementary Fig. 3). To study the propagation of abnormal discharges from one tissue to the other, calcium imaging was performed on both tissues (**b**) and Penicillin G ("Pen") was added to one of the chambers (**c**). **d** Fluorescence pictures of intracellular calcium detected by fluo-4 direct in cerebral tissues at day 45 where one of the chambers (left, "treated") was treated with Penicillin G, whereas the other (right, "untreated") was not. The activity of 522 neurons was detected in the treated (blue circles) and untreated (red circles) tissues and analyzed by live calcium imaging (see also Supplementary Movie 3). Scale bar: 250 μm (n = 6 samples across six independent experiments). **e**, **f** Time traces of four selected cells in the treated (blue) and untreated (red) cerebral tissues pre- (**e**) and post-treatment (**f**) with Penicillin G (see also Supplementary Fig. 4). The black vertical lines indicate instances where all four cells in the treated tissue showed synchronized transient peaks. This synchronicity propagated ~45% of the time to the cells in the untreated tissue. **g**, **h** Quantification of the change in fluorescence intensity (**g**) and fold change in neuronal activity (**h**, log scale) induced by the addition of Penicillin G in the treated (blue) and untreated (red) cerebral tissues. The symbols represent data for each cell; the boxes represent the median, first, and third quartiles; and the whiskers represent the fifth and 95th percentiles of the population data. Asterisk denotes statistically significant difference (Mann–Whitney $U$-test, $p < 10^{-11}$).

reaggregation (Supplementary Fig. 2, 3 and Supplementary Movie 2).

To induce abnormal discharges in one of the tissues, we used the neurotoxic properties of Penicillin G[36], a γ-aminobutyric acid (GABA) A-receptor (GABA$_A$R) blocker of the β-lactam antibiotics family[37], which prevents GABAergic transmission and interferes with the GABA-inhibition and glutamate-excitation equilibrium, causing abnormal electrical discharges[38]. We treated one chamber with Penicillin G by bath application to the culture medium, and monitored the neuronal activity in both chambers using live calcium imaging (Fig. 3b–c). Prior to the Penicillin G treatment, both cerebral tissues showed comparable levels of neuronal activities (Supplementary Fig. 4). Immediately following the Penicillin G treatment, we observed

an increased amount of fluorescence in the treated tissue, compared to the baseline (Fig. 3d, g, Supplementary Fig. 5, Supplementary Movie 3, and Supplementary Data 2). This is accompanied by an increased magnitude and frequency of neuronal activity, as well as synchrony of transient spikes in the treated tissue (Fig. 3d–f and Supplementary Data 3), indicative of abnormal excessive discharges. Subsequently, the neurons in the untreated chamber similarly showed an increased amount of fluorescence intensity and neuronal activity post-treatment (Fig. 3g, h) and synchronicity with neurons in the treated tissue, despite negligible inter-chamber particle diffusion (Supplementary Fig. 6 and Supplementary Data 4). These observations strongly suggest that the immediate changes in the untreated tissue result from discharge propagation through inter-tissue

network connections across the membrane. To our knowledge, this is the first time that signal transmission of abnormal activities (i.e., epileptiform discharge propagation) has been recapitulated in-vitro in 3D.

## Discussion

The engineered cerebral tissues in this study showed characteristics of functional human neuronal networks, including synchronized influxes of extracellular calcium and modular functional connectivity patterns, demonstrating the formation of interconnected networks within the intact tissues. From a developmental perspective, neuronal migration is a key step in the assembly of neuronal circuits and is thought to be necessary for the maturity of interconnected networks in the brain[39]. This step, however, is often inherently suppressed in the generation of 3D in-vitro models, such as brain organoids (e.g., based on a serum-free culture of embryoid body-like aggregates with quick reaggregation or the SFEBq method[40]). As such, the main advantage of the MARC approach, compared to conventional protocols for generating brain organoids, is that it promotes an active (migrative) reaggregation of the cells during network formation, with the support of the matrix (Matrigel). Another potential advantage of the MARC approach is that the active tissue formation allows for manipulations during the migrative reaggregation process to further tune tissue structure and function, which are not possible with the quick reaggregation step in conventional protocols. It is worth noting that the abovementioned reliance on the support of the extracellular matrix (Matrigel) to facilitate 3D reaggregation and tissue formation also makes the approach susceptible to the well-documented weaknesses of Matrigel, especially its batch-to-batch variability[41], which can, in turn, result in variability of the timing of different steps of the reaggregation process. In order to overcome this limitation, it will be useful to explore the use of alternative matrices, such as tunable synthetic hydrogels[41,42]. Moreover, future experiments should shed more light on the mechanisms underlying the cellular processes involved in the reaggregation process and how these are influenced by the physical, mechanical, and biochemical properties of the matrix.

We propose that the matrix-supported reaggregation process in the MARC approach minimizes exogenous (mechanical) perturbations and thereby potentially reduces cellular stresses when compared to the conventional SFEBq methods, which predominantly rely on mechanically enforced embryoid body formations[40]. While a comprehensive characterization of cellular stress is beyond the scope of our present work, as a first test, we examined the expression of the stress marker COPD in cerebral tissues obtained using the MARC approach and in cerebral organoids obtained using SFEBq (commercial STEM-diff protocol, see Methods) after 90 days in 3D culture. Immunostaining results indicate a higher level of COPD expression in the organoids compared to in the MARC-produced cerebral tissues (Supplementary Fig. 7). This preliminary data represents a first indication that our culture method of promoting active reaggregation of cells and spheroids could limit cellular stress and potential adverse effects on tissue functionality[43], which should be further explored in future research. In addition, while the immunohistochemical characterization on major neuronal and neuroglial identities showed comparable cell diversity between the MARC-produced cerebral tissues and cerebral organoids obtained using SFEBq (Supplementary Fig. 1), it will be instructive to examine the cell-type composition of MARC-produced cerebral tissues in further detail using quantitative multi-omics studies.

In the present study, through analysis of calcium imaging data, we observed spatial propagation of excessive discharges between interconnected cerebral tissues, which is a clinically recognized signature of the network propagated epileptic activity in a focal seizure. Specifically, the propagation of abnormal activity in this study was determined in terms of synchronicity between the pharmacologically induced hyper-activated tissue and the untreated tissue, using calcium imaging with a relatively low sampling rate. It will be interesting to extend this further, for example using ultra-fast cameras and microelectrode arrays, to study high-frequency oscillations and wave propagations involved in different phenomena of epileptic seizures. Moreover, the MARC-produced cerebral tissues in this study exhibit a non-organized structure, which is common to all whole-brain approaches. To overcome this limitation for the study of more subtle, complex neurodevelopmental inter-regional anomalies, region-specific approaches[3,15–17] could be implemented in the MARC method.

Taken together, the method introduced in this study to develop 3D neuronal tissues while preserving the potential of organoids opens a range of possibilities for engineering approaches to mechanistically analyze clinically relevant 3D functional network connectivity. The combination of the reaggregation process in the MARC approach and the 3D connection across the membrane in the iS3CC chip facilitates independent treatment of the separated but interconnected tissues, which to our knowledge has not been achieved with existing methods. This enables systematic studies and controlled in-vitro modeling of network pathologies, whereby the activity of one area of the brain can be (pharmacologically) altered and manipulated, which can in turn contribute to drug development.

## Methods

**Cell culture**. Human-induced pluripotent stem cells (hiPSCs) were cultured in a mTeSR medium (STEMCELL Technologies). The cell culture flasks were coated with Matrigel (hECS-qualified matrix, Corning, C354277) diluted in a 1:1 mixture of Dulbecco's Modified Eagle's Medium (DMEM, Gibco) and Ham's F-12 Nutrient Mixture (Gibco) with a v/v ratio of 1:80 for 2 h in an incubator at 37 °C and 5% $CO_2$. On the first day of culture, the hiPSCs were treated with 10 μM ROCK inhibitor Y-27632 (STEMCELL Technologies). The cells were washed using Dulbecco's Phosphate-Buffered Saline (DPBS, Gibco) and the medium was refreshed daily until 80–100% confluence was reached.

**Neural induction**. The cells were switched to the neural induction medium containing 1:1 mixture of N2/B27 medium containing 10 ng/ml basic fibroblast growth factor (b-FGF), 1 μM Dorsomorphin dihydrochloride (Tocris, 3093), 10 μM SB431542 (Tocris, 1614), 100 ng/ml mouse recombinant Sonic Hedgehog (SHH)-C25II (Genscript, Z03050-50), and 10 μM CHIR99021 (Sigma, SML1046). The N2 medium consisted of DMEM/F12 medium (Gibco) with 1× N2 supplement (Gibco, 17502048), 5 μg/ml insulin (Sigma, 19278), 1 mM L-Glutamine (Lonza, 17605E), 100 μM MEM-Non-Essential Amino Acid solution (NEAA) (Gibco), 100 μM 2-mercaptoethanol (Sigma, M3148), and 1:100 Penicillin-Streptomycin (Lonza, 17602E). The B27 medium consisted of a Neurobasal medium (Gibco) and 1× B27 supplement (Gibco, 17504044). The cells were washed daily using DPBS and maintained in the induction medium.

### Supported reaggregation and tissue formation

*Neural differentiation*. After 1 week of neural induction, the cells were dissociated using Accutase (STEMCELL Technologies, 07920) and resuspended in growth factor reduced (GFR) Matrigel (Corning, 734-0269) and neural differentiation medium with a 70:30 v/v ratio at a density of 50,000 cells/chamber. The neural differentiation medium consisted of N2/B27 medium containing 10 ng/ml b-FGF, 20 ng/ml SHH-C25II, and 100 ng/ml human recombinant FGF8 (Gibco, PHG0184). The medium was refreshed daily for 7–10 days as the cells aggregated to form spheroids.

*Pre-terminal differentiation and neural maintenance*. Following spheroid formation, the medium was replaced with a pre-terminal differentiation medium containing a neural differentiation medium without b-FGF. After 10–15 days of culture, during which period the spheroids extended neurites and made extensive connections with each other, the medium was replaced with N2/B27 medium, entering the terminal differentiation phase. From this point, the medium was changed every other day.

**Cerebral organoid formation and maintenance**. Whole-brain organoids were developed using the STEMdiff™ Cerebral Organoid Kit (catalog #08570; StemCell Technologies, Cambridge, Massachusetts, USA) according to the manufacturer's protocols.

**iS3CC chip fabrication**. The three-dimensional model of the device was built in Siemens NX (version NX10) software, from which the model of the negative mold was created. A polycarbonate (PC) negative mold was fabricated using micro-milling (Mikron wf 21 C). A PDMS silicone elastomer kit (Sylgard 184) was used to create the devices using soft lithography. A solution of silicone elastomer and curing agent with a weight ratio of 10:1 was mixed and degassed and then casted into the PC negative molds and cured in the oven at 80 °C for at least 3 h. After that, polyethylene terephthalate (PET) porous membranes with a pore size of 8 μm (ThinCert, 657638) were manually cut into the desired size and placed in the manually created cut in the PDMS at the position as indicated in Supplementary Fig. 2. Protrudes of the membrane from the bottom surface of the PDMS chip were removed using a surgical blade under an upright microscope, to ensure a perpendicular configuration of the membrane when bonded to a thin glass and to avoid membrane folding and partial limitation of the field of view when observing using an inverted microscope. The PDMS chip and the membrane were immobilized on a 0.17 mm glass. We performed this by placing the PDMS chip (containing the membrane) on a spin-coated PDMS pre-polymer (10:1 base to curing agent weight ratio) at 1000 RPM on a 0.17 mm glass slide followed by a curing step of 2 h at 100 °C.

**Cerebral tissue formation and maintenance in the iS3CC chips**. The transparent PDMS devices were repeatedly washed in a biological safety cabinet using 70% ethanol followed by sterilization using UV light in the safety cabinets (3 × 5 min). After the neuronal induction phase, the cells were disassociated using Accutase and resuspended in 35 μl of a mixture of cold GFR-Matrigel and differentiation medium with a 70:30 v/v ratio at a final concentration of 50,000 cells per chamber. The chips containing cells and Matrigel-medium mixture were placed in a Petri dish to prevent contamination (since the chips have an open-top) and kept in an incubator at 37 °C and 5% $CO_2$ for 5 min to polymerize the Matrigel mixture. After that, an additional 200 μl of differentiation medium was added to each chamber and the chips were placed back into the incubator. The medium was refreshed and the culture was continued as described above, with gentler handling to prevent damage to the gel and cells.

**Immunohistochemistry**. The MARC-produced cerebral tissues were fixed in 3.7% paraformaldehyde overnight at 4 °C and washed five times with PBS for 10 min. The MARC-produced cerebral tissues were transferred into 15 and 30% sucrose and allowed to sink overnight at 4 °C, in a sequential manner. After that, OCT compound (Tissue-Tek) embedding medium was added to the MARC-produced cerebral tissues and snap-frozen on liquid nitrogen. Sections of 50 μm were created using cryotome (Microm, HM 550). For immunostaining, the sections were dried at room temperature and subsequently permeabilized for 10 min using 0.5% Triton X-100 in PBS and blocked for 1 h at room temperature using 10% normal donkey serum. After that, sections were incubated in the following primary antibodies diluted in PBS containing 1% normal donkey serum: rabbit polyclonal anti-Sox2 (1:250, Abcam, ab97959), mouse monoclonal anti-beta-Tubulin-III, Tuj1 (1:200, Merck, MAB1637), mouse monoclonal anti-vesicular glutamate transporter-I, VGLUT1 (1:250, Merck, AMAB91041), rat monoclonal anti-Dopamine Transporter, DAT (1:250, Abcam, ab5990), rabbit anti-VGAT (1:100, Merck, AB5062P), mouse anti-GFAP (1:200, Merck, G3893), chicken polyclonal anti-MAP2 (1:500, Abcam, ab5392), rabbit anti-OLIG2 (1:200, Merck, HPA003254) and mouse monoclonal anti-COPD (1:250, Invitrogen, MA5-18287). Fluorescence images of sections were obtained using a confocal microscope (Leica SP8X) with a 10×/0.4 HC PL Apo CS2 or a 40×/0.95 HC PL Apo objective lens, a white light laser (with individually tunable wavelengths 470 nm < λ < 670 nm), and a continuous-wave UV (λ = 405 nm) laser.

**$Ca^{2+}$ imaging**. Live calcium imaging was performed with widefield epifluorescence microscope (Leica DMi8), equipped with temperature, $CO_2$, and humidity control. The tissues were incubated in the recommended concentrations of Fluo-4 direct according to the manufacturer (Molecular Probes, F10471) for 50 min in the incubator at 37 °C and 10 min at room temperature. After that, the tissues were washed five times with the neural maintenance medium. The calcium surges were recorded using an excitation of 488 nm and an emission of 530 nm every 10 s. Fluorescence images were obtained using a widefield epifluorescence microscope (Leica DMi8) equipped with either a 5×/0.15 HC PL Fluotar or a 10×/0.32 HC PL Fluotar objective lens.

**Penicillin treatment**. After 15 min of calcium imaging in normal conditions, the chip was taken out of the live-imaging setup and 170 μl of the total volume of the treated chamber was replaced by a solution of Penicillin G sodium salt (Sigma, 13752) with a concentration of 100 mg/ml (equivalent to 2,8 × 10⁴ IU). The chip was quickly placed back in the same position and the live calcium imaging was continued. This process took ~3 min.

**Analysis of network interconnectivity**. Neuronal activity was analyzed from the time-lapse images (see Supplementary Software 1). To account for possible drifts or deformations of the tissues, the location of the cells was detected in each frame and linked across the frames using the Mosaic particle tracker plugin in ImageJ. The fluorescence intensity of each detected cell was calculated from the image intensity data using MATLAB (version R2018b, The Mathworks Inc.). Occasional gaps in the time traces, due to the cells not being detected in certain frames, were filled using spline interpolation of the intensity values. The background fluorescence in each chamber was calculated in the same way using ten arbitrarily selected cell-sized ROIs in the cell-free regions of each chamber, averaged, and subtracted from the real (cell) data. Further, the intensity values were corrected for imaging artifacts due to out-of-focus fluorescence by subtracting the mean intensity of an annular mask with an outer radius of 18 pixels and an inner radius of 9 pixels (i.e., size of the ROI) for each ROI.

Further analysis of the functional interconnectivity between neurons was performed using a custom-written script in MATLAB. The normalized rate of change in fluorescence ($\Delta F/F$) was calculated using ($F_{cell} - F_{min}$)/$F_{min}$ where $F_{cell}$ is the mean fluorescence of a selected ROI measured in each frame and $F_{min}$ the lowest measured mean fluorescence value of that ROI throughout the imaging window. For the analysis of the neuronal activity upon Penicillin treatment, $\Delta F/F$ was calculated as ($F_{cell} - F_{init}$)/$F_{init}$, where $F_{init}$ is the fluorescence intensity of the ROI at the beginning of the live imaging, to capture the jump in fluorescence intensity due to the addition of Penicillin. Transient spikes with minimum peak prominence larger than twice the magnitude of stochastic noise in the cell-free regions were detected using the "findpeaks" function. To identify synchronized neuronal firings, Pearson's linear correlation coefficient $r$ was computed for each pair of detected ROIs. Following Eguiluz et al.[4], we defined two ROIs to be functionally connected when $r > 0.6$. To further assess the network modularity, we analyzed the functional connectivity using the iterative Louvain community-detection algorithm[33,34] with weighted edges to find the optimal network partitioning. Visualization of the network connectivity was realized using the graph plotting functions in MATLAB.

**Diffusion rate measurements in the iS3CC device**. For analysis of relative diffusion rate, one of the chambers of the iS3CC device was loaded with 100 mg/ml of fluorescein sodium salt in Milli-Q water and the other chamber with pure Milli-Q. At different time points, samples of 5 μl were taken out from the pure Milli-Q chamber and added to a 96 well microplate to a final volume of 100 μl and fluorescent intensity was measured using a plate reader (Synergy HT BioTek Instruments, USA). Based on calibration experiments, the final concentrations in the Milli-Q chamber were measured at time points of 5, 15, 30, and 60 min, as well as 2, 4, 8, 25, and 30 h.

**Statistics and reproducibility**. Cerebral tissues were successfully generated in six independent rounds of the MARC protocol, yielding roughly 120 MARC-produced cerebral tissues. For the immunohistochemistry experiments, for each co-staining group, five MARC-produced cerebral tissues (3–4 sections per tissue) from two independent experimental rounds were used. The calcium imaging and Penicillin-treatment experiments were performed on at least six tissues across three independent experimental rounds. The diffusion experiments were performed on eight iS3CC chips across two independent experimental rounds.

**Reporting summary**. Further information on research design is available in the Nature Research Reporting Summary linked to this article.

## Data availability

Data used to process and plot the results in this study are provided as Supplementary Data 1–4, together with the analysis code provided as Supplementary Software 1. All other data of this study are available from the corresponding authors upon reasonable request.

## Code availability

All codes used in this study are provided as Supplementary Software 1 and make use of MATLAB's (Mathworks) statistics and data analysis toolbox codes.

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

## Acknowledgements

We thank Jan de Boer, Jaap den Toonder, Anton de Louw, and Remco van der Hofstad for insightful discussions, Jurgen Bulsink for help with the fabrication of the iS3CC chip molds, Serena Buscone for support with immunohistochemical staining, Jos Broers and Florence van Tienen (Maastricht University) for sharing the hiPS cells, and Koen Pieterse (ICMS Animation Studio) for help with the illustrations used in this manuscript. The authors acknowledge support from the Dutch Research Council (grant OCENW.XS2.017, to N.A.K.), European Research Council (grant 851960, to N.A.K.), and the Netherlands' Ministry of Education, Culture, and Science (Gravitation program "Materials-Driven Regeneration", grant 024.003.013, to N.A.K. and C.V.C.B.).

## Author contributions

A.S., N.A.K., and C.V.C.B. developed the initial concept; A.S. developed the experimental approach and designed the culture devices, performed experiments, and analyzed data with input from N.A.K. and C.V.C.B.; A.P.A. conceived and supervised the network disorder study; N.A.K. developed and performed the network analysis; A.P.A., N.A.K., and C.V.C.B. provided support and supervised the project; A.S. and N.A.K. wrote the manuscript; All authors discussed results, contributed to the revision of the manuscript, and approved the manuscript.

## Competing interests

The authors declare the following competing interests: A.S., N.A.K., and C.V.C.B. have filed a design right application for the design of the iS3CC chip and a patent application for the MARC culturing protocol. N.A.K. is an Editorial Board Member for *Communications Biology*, but was not involved in the editorial review of, nor the decision to publish this article. The remaining authors declare no competing interests.
