## [Peer Review File · Communications Biology]

Reviewers' comments:

Reviewer #1 (Remarks to the Author):

This paper presents an interesting approach for studying pathologies disrupting the neural networks of the human brain. The concept is based on developing cerebral 3D tissues (spheroids) that are allowed to spontaneously connect via neural outgrowth. The culture protocol to develop the spheroids is also modified from the common practise to first aggregate the initial cell cluster via e.g. hanging-drop method or using round-bottom wells. Further, the authors demonstrate disruptions to neural connections between two spheroids, isolated in a two-chamber PDMS device, when one of the spheroids is treated with Penicillin G.

All in all, the paper includes several new concepts and contributes to the development of both cerebral 3D tissue culture protocols and functional studies. I do appreciate the effort and concept to develop in vitro models of neural network interconnectivity but I do wonder about the in vivo translation potential taken the non-organised structure of the spheroids? Please comment on that. There are e.g. no reports in the paper on brain region specific organisation of the individual spheroids, following the approach of creating in vivo models of the brain using "assembloids".

Also, I have difficulties to fully understand the culture protocol described early in the paper, could it be that the word "therefore" on p. 2 line 47 should read "thereafter"? What are the main advantages of your new method (MARC) compared to e.g. the STEMDiff protocol?

How do you control for only specific activation of the Penicillin to the spheroid in the +ve well? I would expect diffusion of Penicillin also into the -ve chamber, simply due to the concentration gradient, as the spheroids are placed in two static wells. Please describe how you have accounted for this in the experiment. Also, I don't consider the statistics of this study to be sufficient, more repeats should be performed.

Reviewer #2 (Remarks to the Author):

The manuscript presents a modified method for generating brain organoids with neurites outgrowth to recapitulate pathological neural connectivity in a dish. The technique will be of interest to many people in neuroscience; however, more experiments are needed to support the manuscript's central claims.

Specific comments, with recommendations for addressing each comment:

1. The introduction didn't cover the complete picture of the brain organoids and existed technologies. There has been tremendous progress in the field of brain organoids, as well as other systems like 3D brain spheroids, in the last few years. While current brain organoids show neurite ingrowth, several groups reported that stem cell-derived brain spheroids could be generated with neurite outgrowth for studying the propagation of disease pathology and other applications. Please revisit the introduction to cover i) background in brain organoids and other 3D brain systems like spheroids, ii) research gap, and ultimately iii) thesis of the current paper.
2. On page 2, paragraph 2, the authors claimed that "spheroids merged over time, likely through a synapse-mediated migration...". Additional experiments or data are needed to support this significant claim on how spheroids use synapse-mediated migration to form large tissues. To support the claim, the authors need to show synaptic transmission from neuron to neuron or region to region, which induces the polar transition, leading to a change in migration mode and locomotion.
3. On page 3, paragraph 1, and throughout the manuscript, the authors claimed the multiregional tissue patterning of generated cerebral organoids without enough data to support this claim. Additional experiments are needed. First in Fig 1c, most of the staining for various antibodies is present on the tissues' edges. However, DAPI staining shows cells in the middle of the sectioned tissues. Without clear co-staining, it is hard to elaborate on these immunostaining pictures. Second, the co-staining in Fig. S1 is poorly performed and confuses the reader rather than confirming the multiregional tissue assembly. High-quality co-staining with zoom-in is necessary to support the claim of the multiregional structure of the cells.

4. Given the level of detail provided on characterizing the generated organoids, it is challenging for other groups to reproduce the work. From Figure 1C, it looks like that most of the organoids are composed of GABAergic neurons based on immunostaining. It would be interesting and necessary to look further into the composition of the generated organoids in regard to cell types. This would further help other researchers leverage these organoids to understand the mechanisms underlying specific neuropathological or developmental studies.
5. Please provide more details on Supp. Fig. 2 with the step-by-step fabrication of the IS3CC chip since the chip is reported for the first time.
6. On page 9, paragraph 2, the authors claimed that "MARC-produced cerebral tissues hold great potential for uncovering the (patho)physiological features of healthy and diseased neuronal network". There is no data to support this claim. As previously mentioned, several models have already been reported in the 3D spheroids that show neurite outgrowth and can be leveraged to study healthy-disease propagation of specific pathology. The authors need to elaborate further on why their system is required to strengthen the conclusions.
7. Fig. 3 without a straightforward schematic of the timeline for various steps is confusing. It would be beneficial for the figure's readability to add a schematic representing step by step of Pen G addition with timeline followed by signal capturing and fluorescence imaging.
8. More experiments are needed to clarify the diffusion of Penicillin G through the 8 um microporous membrane to the non-treated organoids. Supp. Fig. 5 is supposed to show the diffusion, but it isn't apparent.
9. It would be very helpful if the authors could provide a context on the limitations of the current system in the conclusions.

Reviewer #3 (Remarks to the Author):

The article titled "In-vitro engineered human cerebral tissues mimic pathological circuit disturbances in 3D" shows the development of a multi-chambered tissue culture chip that can be used to interconnect independently constructed cerebral tissues and use that tissue preparation to model propagation of epileptiform discharges. The approach is innovative and generally sound, but at times authors may have exaggerated the results inadvertently. Authors may need to explain the concept and results a little more and provide comparisons with controls, if applicable, to support their claims. Additional experiments with proper controls (conventional organoids etc.) are needed to support the claims. If it can be shown that reaggregated spheroids are very different from conventional spheroids in terms of electrical activity, immunohistochemistry, etc., then the results would be novel and interesting for others in the community.

Additional experiments:

1. Authors may want to add conventional cerebral organoids prepared by methods such as SFEBq in the two chambers to show how reaggregation methodology is better?
2. Authors claim that this is the first time an abnormal activity is propagated in cerebral organoids. However, how the propagation of abnormal activity is different from the propagation of normal activity, and can this preparation be used to study the normal behavior of cerebral organoids. The claim of propagation of abnormal activity was rather confusing.

Manuscript ID: COMMSBIO-21-0031-T

Title: In-vitro engineered human cerebral tissues mimic pathological circuit disturbances in 3D

Reviewer reports and Authors' response

Reviewer 1:

Reviewer's comment:

This paper presents an interesting approach for studying pathologies disrupting the neural networks of the human brain. The concept is based on developing cerebral 3D tissues (spheroids) that are allowed to spontaneously connect via neural outgrowth. The culture protocol to develop the spheroids is also modified from the common practise to first aggregate the initial cell cluster via e.g. hanging-drop method or using round-bottom wells. Further, the authors demonstrate disruptions to neural connections between two spheroids, isolated in a two-chamber PDMS device, when one of the spheroids is treated with Penicillin G.

All in all, the paper includes several new concepts and contributes to the development of both cerebral 3D tissue culture protocols and functional studies. I do appreciate the effort and concept to develop in vitro models of neural network interconnectivity but I do wonder about the in vivo translation potential taken the non-organised structure of the spheroids? Please comment on that. There are e.g. no reports in the paper on brain region specific organisation of the individual spheroids, following the approach of creating in vivo models of the brain using "assembloids".

Authors' response:

We thank the reviewer for the kind words and constructive comments. We are pleased that the innovative approach reported in the manuscript can be readily appreciated.

The reviewer correctly pointed out that, in the original manuscript, we had not extensively discussed the structural organization of the tissues. The reason is that we focused the present study solely on network activity and interconnectivity, which are especially relevant for pathologies that affect the whole brain, such as in the case of an epileptic seizure modeled in this study. For other disease contexts that require more faithful recapitulations of the brain region-specific organization, a possible alternative is indeed the region-specific differentiation approach, followed by the formation of assembloids.

To clarify this point, in the revised manuscript (pages 13-14, lines 275-278) we have included additional comments on brain region-specific protocols and assembloids, including the relevant references:

Moreover, the MARC-produced cerebral tissues in this study exhibit a non-organized structure, which is common to all whole-brain approaches. To overcome this limitation for the study of more subtle, complex neurodevelopmental inter-regional anomalies, region-specific approaches^{3,15-17} could be implemented in the MARC method.

Reviewer's comment:

Also, I have difficulties to fully understand the culture protocol described early in the paper, could it be that the word "therefore" on p. 2 line 47 should read "thereafter"? What are the main advantages of your new method (MARC) compared to e.g. the STEMDiff protocol?

Authors' response:

We thank the reviewer for the careful attention to the wording in our manuscript. We have rephrased the sentence to clarify the protocol and avoid confusion (page 3, lines 61-63):

Thence, neural differentiation took place in a 3D environment for 7 days, accompanied by the rapid formation of spheroids with a size of 200–300 μm (Fig. 1b, Day 7).

We have now also included explanations of the main advantages of the MARC method compared to conventional protocols such as STEMdiff in the revised manuscript:

page 12, lines 235-240

As such, the main advantage of the MARC approach, compared to conventional protocols for generating brain organoids, is that it promotes an active (migrative) reaggregation of the cells during network formation, with the support of the matrix (Matrigel). Another potential advantage of the MARC approach is that the active tissue formation allows for manipulations during the migrative reaggregation process to further tune tissue structure and function, which are not possible with the quick reaggregation step in conventional protocols

page 14, lines 279-284

Taken together, the method introduced in this study to develop 3D neuronal tissues while preserving the potential of organoids opens a range of possibilities for engineering approaches to mechanistically analyze clinically relevant 3D functional network connectivity. The combination of the reaggregation process in the MARC approach and the 3D connection across the membrane in the iS3CC chip facilitates independent treatment of the separated but interconnected tissues, which to our knowledge has not been achieved with existing methods.

In addition to the above advantages, we also propose that the matrix-supported reaggregation process in the MARC approach minimizes exogenous (mechanical) perturbations and thereby potentially reduces cellular stresses when compared to the conventional SFEBq methods, which predominantly rely on mechanically enforced embryoid body formations. While a comprehensive characterization of cellular stress is beyond the scope of our present work, we have nevertheless performed additional experiments where we set up the STEMdiff protocol in our lab and compared the expression of the stress marker COPD in the STEMdiff organoids and MARC tissues at 90 days. Immunohistochemistry stainings indicate that, while the expression levels of neuronal markers were comparable in MARC and STEMdiff samples (Fig. R1 below), COPD is more expressed in STEMdiff organoids than in MARC tissues (Fig. R2 below).

We have included a comment on this point (page 13, lines 259-262) and included these new data in the revised manuscript (Supplementary Fig. 1 and Supplementary Fig. 7) and the associated descriptions of the experiments.

page 13, lines 259-262

This preliminary data represents a first indication of the importance of culture condition on cellular stress and tissue functionality, which should be further explored in future research.

Figure R1 | Multiregional cerebral tissues generated by the MARC method and STEMdiff cerebral organoids. Immunohistochemical co-staining of multiple markers on 50- μ m-thick sections of MARC-produced cerebral tissues (left column) and STEMdiff cerebral organoids (right column) at Day 90 showed no significant difference between the two protocols in the overall expression of distinct neuronal and neuroglial cell types. Co-staining of neural progenitor cells (NPCs; SOX2 in red), early neurons (Tuj1 in green) and mature neurons (MAP2 in blue) (a), co-staining of mature neurons (MAP2 in red), astrocyte marker (GFAP in green) and DNA marker (DAPI in blue) (b), mature oligodendrocytes marker (Olig2 in red), co-staining of mature neurons (MAP2 in green), and DNA marker (DAPI in blue) (c) and co-staining of dopaminergic neurons (DAT in red) and glutamatergic neurons (VGLUT1 in green) and GABAergic neurons (VGAT in blue) (d) (n = 5 samples across 2 independent experiments). Scale bar: 50 μ m.

Figure R2. Expression of the stress marker in MARC-produced cerebral tissues and STEMdiff cerebral organoids. Immunohistochemical staining of the stress marker (COPD) in MARC-produced cerebral tissues (a) and STEMdiff cerebral organoids (b). The results indicate a higher level of COPD expression in the organoids compared to in the MARC-produced cerebral tissues. The right panel shows the zoom-in of the image in the left panel as indicated by the white squares (n = 5 samples across 2 independent experiments). Scale bar: 500 μ m for the left panel and 50 μ m for the right panel.

Reviewer's comment:

How do you control for only specific activation of the Penicillin to the spheroid in the +ve well? I would expect diffusion of Penicillin also into the -ve chamber, simply due to the concentration gradient, as the spheroids are placed in two static wells. Please describe how you have accounted for this in the experiment. Also, I don't consider the statistics of this study to be sufficient, more repeats should be performed.

Authors' response:

We agree with the reviewer that the possible diffusion of Penicillin between the chambers should be accounted for. In the original manuscript, we showed in an experiment using fluorescently tagged molecules that, within the timeframe of the Penicillin experiment (< 1 h), negligible diffusion across the membrane was detected. To further confirm this, we have performed additional experiments with 6 additional iS3CC chips using fluorescein sodium salt (with a similar size and the same concentration as those used in the Penicillin experiment). The fluorescence intensity of the solution in the chamber initially containing pure MilliQ (the receiving chamber) was measured over time using a plate reader and converted to concentration using a calibration curve. As shown in Figure R3 below, the recorded concentration in the receiving chamber in the first 60 min remains negligible (between 0-1 μ g/ml) and increases only after several hours. In contrast, the propagation (synchronicity) of the discharges in the MARC tissues across the membrane (Figure 3 in the manuscript) starts immediately after addition of Penicillin. It should also be noted that, whereas diffusion of the fluorescein sodium salt takes place in pure MilliQ water and across a bare membrane in the diffusion experiment, the diffusion of Penicillin sodium salt takes place in Matrigel and across a membrane covered by MARC tissues in the Penicillin-treatment experiment. Thus, we expect the former to be a gross overestimation of the diffusion rate in

the Penicillin-treatment experiments. Given the negligible diffusion within the timeframe of the Penicillin-treatment experiments (60 min), we believe that the observed propagation of the abnormal discharges occurs through the neuronal transmissions between the tissues.

Figure R3. | Measurement of particle transfer between the chambers of the iS3CC chip across the porous membrane. a, To simulate any potential diffusion between the two chambers, fluorescein sodium salt with comparable molecular weight (376.27 g/mol) to the Penicillin G sodium salt (367.37 g/mol) used in Figure 3 of the main text was added to one of the chambers of the iS3CC with the same final concentration as Penicillin treatment (100 mg/ml), whereas MilliQ water was added to the other chamber. The fluorescence of the solution in the latter chamber was measured over time using a plate reader and converted to concentration using a calibration curve. b, The measured concentration as a function of time (mean \pm standard deviation, $n = 6$). Inset shows zoom-in of the first hour, i.e., the time-frame of the *in vitro* seizure experiment in Figure 3. Error bars in the first 3 data points are smaller than the symbols. The data shows negligible diffusion across the membrane within this experimental time-frame. Furthermore, the diffusion of the fluorescent sodium salt takes place in pure MilliQ water and across a bare membrane in this diffusion experiment; the diffusion of Penicillin takes place in Matrigel and across a membrane covered by MARC tissues in the seizure experiment. Thus, we expect the former to be a gross overestimation of the diffusion rate in the Penicillin-treatment experiments. Given the negligible diffused concentrations within the experimental timeframe (60 min) of the Penicillin-treatment experiment, we believe that the observed propagation of the abnormal discharges occurs through the neuronal transmissions between the tissues.

We have now included Figure R3 as Supplementary Fig. 6 in the revised Supplementary Information and clarified the experimental details in the figure caption.

To address the reviewer's concern regarding experimental repeats, during the revision we have conducted 2 additional full rounds of MARC tissue generation, characterizations, and measurements. In total, we have formed roughly 120 MARC tissues (40 during the revision) across 6 independent experiments (2 during the revision). We have now included a section on Statistics and Reproducibility on page 23 lines 533-539 covering this point:

Statistics and Reproducibility. Cerebral tissues were successfully generated in 6 independent rounds of the MARC protocol, yielding roughly 120 MARC tissues. For the immunohistochemistry experiments, for each co-staining group, 5 MARC tissues (3-4 sections per tissue) from 2 independent experimental rounds were used. The calcium imaging and Penicillin-treatment experiments were performed on at least 6 tissues across 3 independent experimental rounds. The diffusion experiments were performed on 8 iS3CC chips across 2 independent experimental rounds.

Reviewer 2:

Reviewer's comment:

The manuscript presents a modified method for generating brain organoids with neurites outgrowth to recapitulate pathological neural connectivity in a dish. The technique will be of interest to many people in neuroscience; however, more experiments are needed to support the manuscript's central claims.

Authors' response:

We are pleased that the usefulness of the technique to the community can be readily appreciated and we thank the reviewer for the helpful remarks and suggestions. We have addressed all of the reviewer's comments, performed additional experiments, and adjusted the manuscript accordingly, as detailed below.

Reviewer's comment:

1. The introduction didn't cover the complete picture of the brain organoids and existed technologies. There has been tremendous progress in the field of brain organoids, as well as other systems like 3D brain spheroids, in the last few years. While current brain organoids show neurite ingrowth, several groups reported that stem cell-derived brain spheroids could be generated with neurite outgrowth for studying the propagation of disease pathology and other applications. Please revisit the introduction to cover i) background in brain organoids and other 3D brain systems like spheroids, ii) research gap, and ultimately iii) thesis of the current paper.

Authors' response:

We agree with the reviewer that a clearer elaboration of the state of the art, the scientific gap, and the thesis of the current study helps in clarifying the advances that our study brings. We have adjusted the introduction part of the manuscript, included the relevant references, and clarified the focus of our work on network activity and interconnectivity in the revised manuscript (page 2, lines 26-43):

Three-dimensional (3D) neuronal models, such as brain organoids, combined with recent advances in high-resolution 3D imaging techniques, gene editing, single-cell omics, and patient-derived induced pluripotent stem cells (iPSCs), have provided pioneering platforms for understanding various aspects of brain development¹⁻⁵ and brain pathologies⁶⁻¹⁴. Furthermore, fused region-specific organoids and assembloids^{3,15-17} have been successfully developed to recapitulate in-vivo inter-regional and inter-cellular interactions in 3D. However, despite this exciting progress, building a manipulatable in-vitro model to study the altered or disrupted 3D functional interconnectivity in multiregional network pathologies such as a focal epileptic seizure remains a major challenge^{18,19}.

Efforts have been made to create engineered platforms for studying interconnectivity between 3D neuronal cell cultures (e.g., using interconnected spheroids²⁰⁻²³). However, these approaches lack either the 3D connectivity between the interconnected co-cultures (since the connections are guided through micro channels) or the cellular diversity and complex functionality of organoid approaches (for an overview, see recent reviews by Brofiga et al.¹⁸ and Park et al.¹⁹). There is a clear need for a new approach to develop neuronal tissue models that retain the in-vivo biomimicry potential of organoids while presenting the possibility of spatial control of the tissue configuration in a well-defined engineered culture platform that allows 3D connectivity^{19,24}.

Reviewer's comment:

2. On page 2, paragraph 2, the authors claimed that "spheroids merged over time, likely through a synapse-mediated migration...". Additional experiments or data are needed to support this significant claim on how spheroids use synapse-mediated migration to form large tissues. To support the claim, the authors need to show synaptic transmission from neuron to neuron or region to region, which induces the polar transition, leading to a change in migration mode and locomotion.

Authors' response:

We thank the reviewer for bringing up this point. We indeed observed that the MARC method led to spontaneous merging of spheroids and our initial hypothesis that the merging occurred through synapse-mediated migration was based on *in-vivo* observations of Ohtaka-Maruyama et al. (Science 360:313-317, 2018). As suggested by the reviewer, we performed additional experiments to check this hypothesis. As a relatively simple test, we added the sodium channel blocker Tetrodotoxin (TTX), which prevents the generation and propagation of excitatory action potentials, to the cells at day 0 in 3D for 25 days. We found that TTX-treated cells in suspension reaggregated similarly to those in the untreated MARC samples (Figure R4 below). This finding disproves the hypothesis of synapse-mediated migration during spheroid merging. More systematic experiments in a dedicated study are needed to uncover the mechanism behind the reaggregation process.

We have now removed this statement from the text and commented on this open question in the Discussion (page 12, lines 246-249):

Moreover, future experiments should shed more light on the underlying mechanisms behind the cellular processes involved in the reaggregation process and how these are influenced by the physical, mechanical, and biochemical properties of the matrix.

Figure R4. In order to investigate whether synaptic activities affect the cellular reaggregation during the MARC protocol, we used the sodium channel blocker TTX, which prevents the generation and propagation of excitatory action potentials. The cells were treated with 2 μ M TTX from day 0 to day 25 in 3D. TTX-treated cells in suspension reaggregated (right panel) similarly to those in the untreated MARC samples (left panel). Scale bar: 500 μ m.

Reviewer's comment:

3. On page 3, paragraph 1, and throughout the manuscript, the authors claimed the multiregional tissue patterning of generated cerebral organoids without enough data to support this claim. Additional experiments are needed. First in Fig 1c, most of the staining for various antibodies is present on the tissues' edges. However, DAPI staining shows cells in the middle of the sectioned tissues. Without clear co-staining, it is hard to elaborate on these immunostaining pictures. Second, the co-staining in Fig. S1 is poorly performed and confuses the reader rather than confirming the multiregional tissue assembly. High-quality co-staining with zoom-in is necessary to support the claim of the multiregional structure of the cells.

4. Given the level of detail provided on characterizing the generated organoids, it is challenging for other groups to reproduce the work. From Figure 1C, it looks like that most of the organoids are composed of GABAergic neurons based on immunostaining. It would be interesting and necessary to look further into the composition of the generated organoids in regard to cell types. This would further help other researchers leverage these organoids to understand the mechanisms underlying specific neuropathological or developmental studies.

Authors' response:

We thank the reviewer for pointing out the limited quality of the immunostaining images in the original manuscript. We have now optimized the sample preparation steps and re-performed the co-staining experiments with the optimized parameters. The results, including high-resolution images and zoom-ins, are summarized in Figure R1 above (in the response to Reviewer 1) and Figure R5 below. The high-resolution images show the presence of different neuronal and neuroglial cell types comparable to those found in multiregional whole-brain organoids obtained using conventional protocols such as STEMdiff.

It is important to add that, for whole-brain differentiation approaches where the regionalization of the markers is not controlled (as opposed to fused organoids approaches/assembloids), it is very common to see significant tissue-to-tissue/organoid-to-organoid variability in both marker expression and the location of the cells. An example can be found in Extended Data Figure 1, panel b from Quadrato et al. (Nature 545:48-53, 2017), where different markers from different brain regions at different time points are expressed irregularly at different locations (either at the edge or the center of the organoids). Our result is consistent with this observation.

Finally, we agree with the reviewer that it will be interesting to examine in further detail the cell type composition of the generated MARC tissues in future research using quantitative multi-omics studies. To address this point, we have added a short comment in the manuscript (page 13, lines 262-266):

In addition, while the immunohistochemical characterization on major neuronal and neuroglial identities showed comparable cell diversity between the MARC-produced cerebral tissues and cerebral organoids obtained using SFEBq (Supplementary Fig. 1), it will be instructive to examine the cell type composition of MARC-produced cerebral tissues in further detail using quantitative multi-omics studies.

Figure R5 is now included as Figure 1c and 1d in the revised manuscript, whereas Figure R1 is now included as Supplementary Fig. 1 in the revised Supplementary Information.

Figure R5. Immunohistochemical co-staining of cryosections of MARC-produced cerebral tissues at Day 90 revealed the presence of markers of neural progenitor cells (NPCs; SOX2), early and mature neurons (Tuj1 and MAP2), mature excitatory Glutamatergic neurons (VGLUT1), inhibitory GABAergic neurons (VGAT), mature dopaminergic neurons (DAT), and astrocytes (GFAP), indicating the cellular diversity of the MARC-produced cerebral tissues (n = 5 samples across 2 independent experiments). Scale bar: 500 μ m. The rightmost column shows zoom-ins of the images as indicated by the white squares. Scale bar: 100 μ m.

Reviewer's comment:

5. Please provide more details on Supp. Fig. 2 with the step-by-step fabrication of the iS3CC chip since the chip is reported for the first time.

Authors' response:

We appreciate the reviewer's suggestion to add more detail on the fabrication process of the iS3CC chip. The iS3CC chip was fabricated using lithography techniques common in PDMS device fabrication and organ-on-chip models. The level of fabrication detail provided is sufficient to instruct a researcher knowledgeable in device fabrication and is comparable to that commonly provided for the fabrication of other organ-on-chip models (see, for example, Benam et al. Nature Methods 13:151-157, 2016 and Nashimoto et al. Biomaterials 229:119547, 2020). Nevertheless, we would like to ensure that the features of the chip can be understood by the reader. As such, we have added additional information on the fabrication steps of the iS3CC chip (page 20, lines 437-452). In addition, we provided more details in the caption of the Supplementary Fig. 2. We trust that the provided details have further clarified the fabrication process.

page 20, lines 437-452

The three-dimensional model of the device was built in Siemens NX (version NX10) software, from which the model of the negative mold was created. A polycarbonate (PC) negative mold was fabricated using micro-milling (Mikron wf 21C). A PDMS silicon elastomer kit (Sylgard 184) was used to create the devices using soft lithography. A solution of silicon elastomer and curing agent with a weight ratio of 10:1 was mixed and degassed and then casted into the PC negative molds and cured in the oven at 80 °C for at least 3 hours. After that, polyethylene terephthalate (PET) porous membranes with a pore size of 8 μm (ThinCert, 657638) were manually cut into the desired size and placed in the manually created cut in the PDMS at the position as indicated in Supplementary Fig. 2. Protrudes of the membrane from the bottom surface of the PDMS chip were removed using a surgical blade under an upright microscope, to ensure a perpendicular configuration of the membrane when bonded to a thin glass and to avoid membrane folding and partial limitation of field of view when observing using an inverted microscope. The PDMS chip and the membrane were immobilized on a 0.17 mm glass. We performed this by placing the PDMS chip (containing the membrane) on a spin-coated PDMS pre-polymer (10:1 base to curing agent weight ratio) at 1000 RPM on a 0.17 mm glass slide followed by a curing step of 2 hours at 100 °C.

Reviewer's comment:

6. On page 9, paragraph 2, the authors claimed that "MARC-produced cerebral tissues hold great potential for uncovering the (patho)physiological features of healthy and diseased neuronal network". There is no data to support this claim. As previously mentioned, several models have already been reported in the 3D spheroids that show neurite outgrowth and can be leveraged to study healthy-disease propagation of specific pathology. The authors need to elaborate further on why their system is required to strengthen the conclusions.

Authors' response:

The reviewer is correct that there have been efforts by other researchers for growing 3D spheroids with neurite (bundle) outgrowth. Yet, we believe that the combination of the reaggregation process in the MARC approach and the possibility of systematically studying the three-dimensional neuronal network interconnectivity through the use of the iS3CC chip is unique and ideally suited for dissecting the (patho)physiological features of healthy and diseased neuronal networks. This approach enables three-dimensional connection and independent treatment of the interconnected tissues (see also our response to point #8 below) that, to our knowledge, has not been possible so far with any other approach. For example, in the present study, we demonstrate this concept to mimic a pathological focal seizure,

whereby the activity of one area of the brain is (pharmacologically) altered. As such, our combination of the MARC approach and the iS3CC chip opens the possibility to explore the cellular network origins of a wide range of clinical pathologies, such as altered interconnectivity due to neurodegeneration or region-specific interactions.

To clarify this point, we have added additional comments in the revised manuscript:

page 2, lines 35-43

Efforts have been made to create engineered platforms for studying interconnectivity between 3D neuronal cell cultures (e.g., using interconnected spheroids²⁰⁻²³). However, these approaches lack either the 3D connectivity between the interconnected co-cultures (since the connections are guided through micro channels) or the cellular diversity and complex functionality of organoid approaches (for an overview, see recent reviews by Brofiga et al.¹⁸ and Park et al.¹⁹). There is a clear need for a new approach to develop neuronal tissue models that retain the in-vivo biomimicry potential of organoids while presenting the possibility of spatial control of the tissue configuration in a well-defined engineered culture platform that allows 3D connectivity^{19,24}.

page 14, lines 279-287

Taken together, the method introduced in this study to develop 3D neuronal tissues while preserving the potential of organoids opens a range of possibilities for engineering approaches to mechanistically analyze clinically relevant 3D functional network connectivity. The combination of the reaggregation process in the MARC approach and the 3D connection across the membrane in the iS3CC chip facilitates independent treatment of the separated but interconnected tissues, which to our knowledge has not been achieved with existing methods. This enables systematic studies and controlled in-vitro modeling of network pathologies, whereby the activity of one area of the brain can be (pharmacologically) altered and manipulated, which can in turn contribute to drug development.

Reviewer's comment:

7. Fig. 3 without a straightforward schematic of the timeline for various steps is confusing. It would be beneficial for the figure's readability to add a schematic representing step by step of Pen G addition with timeline followed by signal capturing and fluorescence imaging.

Authors' response:

We have updated the figure and added schematics representing the sequential steps involved in this experiment. Figure 3 in the manuscript (Figure R6 below) now appears as follows:

Figure R6 | Signal propagation between interconnected cerebral tissues. a-c, Schematics indicating the steps involved in the Penicillin-treatment experiments. Two cerebral tissues were generated in the two chambers of the PDMS-based iS3CC chip (see also Supplementary Fig. 2 for more details of the chip) and formed a connection through the porous membrane (a, see also Supplementary Fig. 3). To study the propagation of abnormal discharges from one tissue to the other, calcium imaging was performed on both tissues (b) and Penicillin G (“Pen”) was added to one of the chambers (c). d, Fluorescence pictures of intracellular calcium detected by fluo-4 direct in cerebral tissues at day 45 where one of the chambers (left, “treated”) was treated with Penicillin G, whereas the other (right, “untreated”) was not. The activity of 522 neurons was detected in the treated (blue circles) and untreated (red circles) tissues and analyzed by live calcium imaging (see also Supplementary Movie 3). Scale bar: 250 μm ($n = 6$ samples across 6 independent experiments). e-f, Time traces of 4 selected cells in the treated (blue) and untreated (red) cerebral tissues pre- (e) and post-treatment (f) with Penicillin G (see also Supplementary Fig. 4). The black vertical lines indicate instances where all 4 cells in the treated tissue showed synchronized transient peaks. This synchronicity propagated $\sim 45\%$ of the time to the cells in the untreated tissue. g-h, Quantification of the change in fluorescence intensity (g) and fold change in neuronal activity (h, log scale) induced by addition of Penicillin G in the treated (blue) and untreated (red) cerebral tissues. The symbols represent data for each cell; the boxes represent the median, 1st and 3rd quartiles; and the whiskers represent the 5th and 95th percentiles of the population data. Asterisk denotes statistically significant difference (Mann-Whitney U test, $p < 10^{-11}$).

Reviewer's comment:

8. More experiments are needed to clarify the diffusion of Penicillin G through the 8 um microporous membrane to the non-treated organoids. Supp. Fig. 5 is supposed to show the diffusion, but it isn't apparent.

Authors' response:

Supplementary Figure 5 in the original manuscript showed an experiment whereby we simulated the potential diffusion of Penicillin sodium salt across the membrane using fluorescently tagged molecules. Within the experimental timeframe of the Penicillin-treatment experiment (< 1 h), negligible diffusion across the membrane could be detected. To further confirm this, we have performed additional experiments with 6 additional iS3CC chips using fluorescein sodium salt (with a similar size and the same concentration as those used in the Penicillin experiment). The fluorescence of the solution in the chamber initially containing pure MilliQ (the receiving chamber), was measured over time using a plate reader and converted to concentration using a calibration curve. As shown in Figure R7 below, the recorded concentrations in the receiving chamber in the first 60 min remains negligible (between 0-1 µg/ml) and increases only after several hours. In contrast, the propagation (synchronicity) of the discharges in the MARC tissues across the membrane (Figure 3 in the manuscript) starts immediately after addition of Penicillin. It should also be noted that, whereas diffusion of the fluorescein sodium salt takes place in pure MilliQ water and across a bare membrane in the diffusion experiment, the diffusion of Penicillin takes place in Matrigel and across a membrane covered by MARC tissues in the Penicillin-treatment experiment. Thus, we expect the former to be a gross overestimation of the diffusion rate in the Penicillin-treatment experiments. Given the negligible diffusion within the timeframe of the Penicillin-treatment experiments (60 min), we believe that the observed propagation of the abnormal discharges occurs through the neuronal transmissions between the tissues.

We have now included Figure R7 as Supplementary Fig. 6 in the revised Supplementary Information and clarified the experimental details in the figure caption.

Figure R7. | Measurement of particle transfer between the chambers of the iS3CC chip across the porous membrane. a, To simulate any potential diffusion between the two chambers, fluorescein sodium salt with comparable molecular weight (376.27 g/mol) to the Penicillin G sodium salt (367.37 g/mol) used in Figure 3 of the main text was added to one of the chambers of the iS3CC with the same final concentration as Penicillin treatment (100 mg/ml), whereas MilliQ water was added to the other chamber. The fluorescence of the solution in the latter chamber was measured over time using a plate reader and converted to concentration using a calibration curve. b, The measured concentration as a function of time (mean \pm standard deviation, $n = 6$). Inset shows zoom-in of the first hour, i.e., the time-frame of the Penicillin-treatment experiment in Figure 3. Error bars in the first 3 data points are smaller than the symbols. The data shows negligible diffusion across the membrane within this experimental time-frame. Furthermore, the diffusion of the fluorescent sodium salt takes place in pure MilliQ water and across a bare membrane in this diffusion experiment; the diffusion of Penicillin takes place in Matrigel and across a membrane covered by MARC tissues in the seizure experiment. Thus, we expect the former to be a gross overestimation of the diffusion rate in the Penicillin-treatment experiments. Given the negligible diffused concentrations within the experimental timeframe (60 min) of the Penicillin-treatment experiment, we believe that the observed propagation of the abnormal discharges occurs through the neuronal transmissions between the tissues.

Reviewer's comment:

9. It would be very helpful if the authors could provide a context on the limitations of the current system in the conclusions.

Authors' response:

The main limitation of the MARC approach is its dependence on the support of an extracellular matrix (Matrigel in this study), which is widely known to be prone to batch-to-batch differences (see, for example, Aisenbrey and Murphy, Nature Reviews Materials 5:539-551, 2020) and as such can result in variability of the timing of different steps of the reaggregation process. Another is the non-organized structure of the MARC tissues, which is a general limitation for all whole-brain approaches. We have now discussed these current limitations in the revised manuscript:

page 12, lines 240-246

It is worth noting that the abovementioned reliance on the support of the extracellular matrix (Matrigel) to facilitate 3D reaggregation and tissue formation also makes the approach susceptible to the well-documented weaknesses of Matrigel, especially its batch-to-batch variability⁴², which can in turn result in variability of the timing of different steps of the

reaggregation process. In order to overcome this limitation, it will be useful to explore the use of alternative matrices, such as tunable synthetic hydrogels^{42,43}.

pages 13-14, lines 275-278

Moreover, the MARC-produced cerebral tissues in this study exhibit a non-organized structure, which is common to all whole-brain approaches. To overcome this limitation for the study of more subtle, complex neurodevelopmental inter-regional anomalies, region-specific approaches^{3,15-17} could be implemented in the MARC method.

Reviewer 3:

Reviewer's comment:

The article titled "In-vitro engineered human cerebral tissues mimic pathological circuit disturbances in 3D" shows the development of a multi-chambered tissue culture chip that can be used to interconnect independently constructed cerebral tissues and use that tissue preparation to model propagation of epileptiform discharges. The approach is innovative and generally sound, but at times authors may have exaggerated the results inadvertently. Authors may need to explain the concept and results a little more and provide comparisons with controls, if applicable, to support their claims. Additional experiments with proper controls (conventional organoids etc.) are needed to support the claims. If it can be shown that reaggregated spheroids are very different from conventional spheroids in terms of electrical activity, immunohistochemistry, etc., then the results would be novel and interesting for others in the community.

Authors' response:

We thank the reviewer for the kind words and constructive comments. We have performed additional experiments, including setting up a conventional organoid protocol in our lab, and revised our manuscript to address the reviewer's remarks and suggestions, as described in detail below, and we are pleased that these have indeed strengthened our manuscript.

Reviewer's comment:

1. Authors may want to add conventional cerebral organoids prepared by methods such as SFEBq in the two chambers to show how reaggregation methodology is better?

Authors' response:

We thank the reviewer for the suggestion to make a direct comparison with conventional methods. In order to address this point, we set up the commercially available STEMdiff protocol (based on Lancaster et al. Nature 501:373–379, 2013) in our lab. A direct comparison between the tissues obtained using the MARC method and the organoids obtained using STEMdiff showed similar cell-type distribution, but a higher expression of a cellular stress marker (COPD) in the latter (see our response to Reviewer 1's second comment above). Moreover, as suggested by the reviewer, we added cerebral organoids obtained using the STEMdiff into the i3CC chips in 3D culture ($n = 6$). As shown in Figure R7 below, even after 3 weeks of culture, the STEMdiff organoids showed no sign of connection across the membrane and in some cases dissociation was visible. These results illustrate the advantage of the MARC method for modeling network disorders such as epilepsy in the i3CC chip, where connection across the membrane is needed.

Figure R7. STEMdiff cerebral organoids were added to the chambers of the iS3CC chips in the maturation phase after 25 days (top row). At day 45 (bottom row), the STEMdiff organoids showed no sign of connection across the membrane and in some cases dissociation was visible. The white dashed lines indicate the porous membrane. Scale bar: 500 μm .

Reviewer's comment:

2. Authors claim that this is the first time an abnormal activity is propagated in cerebral organoids. However, how the propagation of abnormal activity is different from the propagation of normal activity, and can this preparation be used to study the normal behavior of cerebral organoids. The claim of propagation of abnormal activity was rather confusing.

Authors' response:

We thank the reviewer for pointing out that the definition of propagation of normal and abnormal activities was not immediately clear. From an experimental perspective, the neural network analysis of microscopy techniques, such as calcium imaging (i.e., with low sampling rates) that was used in our present study, relies solely on the analysis of synchronous activities (Verstraelen et al. *Frontiers in Neuroscience* 12:389, 2018), which does not allow for studying the normal propagation of complex oscillatory waves. Such examination can only be done using either ultrafast cameras or electrophysiology technologies such as micro electrode arrays. This technology has not been available to date, but is currently being developed in our laboratory.

In the present study, the tissues are physically connected across the membrane whereby the membrane allows for independent chemical manipulation, in this case Penicillin. Penicillin is a known inducer of excessive abnormal and synchronous activities (Avoli and Jefferys, *Journal of Neuroscience Methods* 260:26-32, 2016) and was here shown to affect the "treated" tissue, which furthermore also exhibited synchronicity with neurons of the "untreated" tissue (Figure 3 in the manuscript). This synchronicity of the neurons in the focus of a focal epileptic seizure with neurons in other areas is clinically termed as propagation of abnormal activities. We have now included additional explanation to clarify this point and to mention potential future studies (page 13, lines 267-275):

In the present study, through analysis of calcium imaging data, we observed spatial propagation of excessive discharges between interconnected cerebral tissues, which is a clinically recognized signature of network propagated epileptic activity in a focal seizure.

Specifically, the propagation of abnormal activity in this study was determined in terms of synchronicity between the pharmacologically induced hyper-activated tissue and the untreated tissue, using calcium imaging with a relatively low sampling rate. It will be interesting to extend this further, for example using ultra-fast cameras and microelectrode arrays, to study high-frequency oscillations and wave propagations involved in different phenomena of epileptic seizures.

REVIEWERS' COMMENTS:

Reviewer #1 (Remarks to the Author):

The authors have addressed all my concerns adequately and I would be happy to see this manuscript published.

Reviewer #2 (Remarks to the Author):

The authors have addressed my major concerns.